# Systematic evaluation of medication adherence determinants across 137 active substances on population-level real-world health data

Kerli Mooses [1] ✉, Marek Oja[1], Maria Malk[1], Helene Loorents[1], Maarja Pajusalu[1], Nikita Umov[2], Sirli Tamm [1], Johannes Holm[1], Hanna Keidong[1], Taavi Tillmann[2], Sulev Reisberg [1], Jaak Vilo [1] & Raivo Kolde[1]

## Abstract

**Background** The current knowledge about medication adherence is based on studies focusing only on few health conditions and little is known about how strongly adherence is shaped by person-specific behaviour. The aim of the cohort study is to 1) evaluate the effect of multiple factors affecting medication adherence in a consistent manner across 137 active substances, and 2) calculate individual medication adherence score (IMAS), evaluate its predictive power, stability over time, and impact on health outcomes. In essence, IMAS describes persons' medication-taking "baseline".

**Methods** We utilised a representative dataset with electronic health records, claims, and dispensed medications across 137 active substances and applied continuous multiple interval measures of medication availability (CMA). To assess the effect of various demographic, health, and medication-related variables on CMA, we employed linear mixed models.

**Results** Here we show that the medication adherence ranged from 0.423 (albuterol, 95% CI 0.414–0.432) to 0.922 (warfarin, 95% CI 0.917–0.926). The demographic, health- and medication-related factors explained 11.6% and IMAS 22.0% of the variation in adherence. IMAS predicted adherence across medication classes, reduced the risk of overall hospitalisation (hazard ratio = 0.76, 95% CI 0.60–0.97, p < 0.05) and cause-specific incidence for 17 conditions.

**Conclusions** Thus, IMAS represents a person-level metric that captures baseline medication-taking behaviour across therapeutic classes and predicts both medication adherence as well as health outcomes. Our analysis suggests that medication-taking behaviour represents a broader patient-level phenomenon manifesting consistently across medications, suggesting its potential for personalised interventions in clinical practice and more efficient public health strategies and policies.

## Plain language summary

Taking medications as prescribed is important, especially for chronic diseases, to delay disease progression. However, many people struggle to do so and thus, have low medication adherence. To date, little is known about how strongly adherence is shaped by person-specific behaviour. In our study, we developed a the Individual Medication Adherence Score (IMAS), which reflects a person's usual tendency to follow their pre-scriptions. We found that this tendency is stable over time and can predict health outcomes. Knowing a patient's IMAS could help doctors identify who might need extra support to take their medications regularly.

Adherence to medications is crucial for achieving better health and clinical outcomes[1–3], which in turn reduces hospitalisation and annual healthcare costs[3,4]. Moreover, a 16–21% reduction in mortality risk has been reported for patients with good adherence[1,5]. Medication adherence is a complex multifactorial phenomenon[6,7]. A review of systematic reviews identified more than 800 individual, medication-related and healthcare system-related factors potentially influencing medication adherence[7]. However, the effect size, significance and even direction were found to be inconsistent across studies[7]. Thus, more research is needed on factors influencing medication adherence.

[1]Institute of Computer Science, University of Tartu, Tartu, Estonia. [2]Institute of Family Medicine and Public Health, University of Tartu, Tartu, Estonia. ✉e-mail: kerli.mooses@ut.ee

One challenge of synthesising the existing knowledge lies in the nature of studies that mostly focus on one or a few selected chronic conditions[1,4,8]. As a result, little is known about the adherence patterns within the same individual for co-prescribed medications. Considering that the prevalence of polypharmacy is 55% among people older than 65 years[9], broadening the analysis with a wide range of medications for chronic conditions is necessary. Moreover, to date, there is a gap in understanding the extent to which overall adherence is shaped by medication-specific factors, as opposed to person-specific traits or different life events. Therefore, a qualitatively different approach is needed to disentangle the complex network of clinical, socio-economic and behavioural factors influencing medication adherence.

The emergence of population-level prescription databases, linked with patient medical records, allows to systematically and with consistent methodology evaluate the influence of different factors on medication adherence while distinguishing medication-level, person-specific and time-dependent effects. In the current study, we take advantage of a representative random sample of the Estonian population to study the determinants of medication adherence over a hundred active substances prescribed for long-term use. We calculate individual medication adherence score (IMAS) using a wide range of active substances prescribed for chronic conditions and apply IMAS in further analysis. Conceptually, IMAS captures an individual's inherent medication-taking behaviour—their personal "baseline" adherence tendency that persists across prescriptions and time. We set out to first compare the effect sizes of multiple factors affecting medication adherence calculated in a consistent manner across 137 active substances. Secondly, we calculate IMAS using information about 137 active substances, describe its predictive power and stability over time, and evaluate its effect on health outcomes. We hypothesise that persons' medication-taking habit can be predicted based on previously administered medications, and it remains similar despite the administered medication, age, and treated disease.

Our analysis shows that demographic, health- and medication-related factors explain only a modest proportion of adherence variation, whereas an IMAS accounts for a notably larger share. IMAS consistently predicts adherence across medication classes and shows moderate stability over time.

Higher IMAS is associated with better future health outcomes, including a reduced risk of overall hospitalisation and lower incidence of multiple chronic conditions.

## Methods

### Data and setting

The dataset consisted of all electronic health records (EHR), healthcare service provision claims, and prescribed medications from 01.01.2012 to 31.12.2019 for randomly selected 10% of the Estonian population ($N = 150,824$). The EHR, claims and prescriptions data originated from three health databases with national coverage, storing information from almost all healthcare settings (hospitals, specialists, family doctors, labs, pharmacies). EHR covers data from all private and state-owned healthcare providers regardless of the insurance of the patient, while claims covers only people with public health insurance (approximately 95% of the Estonian population[10]). The drug prescription database includes all prescribed drugs[11]— their active substance, Anatomical Therapeutic Chemical (ATC) code, product name and code, amount, administration guidelines, purchase date and location, a healthcare provider who issued the medication, and the International Statistical Classification of Diseases and Related Health Problems 10th Revision (ICD-10) code for the condition being treated. The databases were linked using a unique personal ID code given to all Estonian residents, and data were transferred to the Observational Medical Outcomes Partnership (OMOP) common data model (CDM) version 5.4[12]. The overall workflow of the analysis is presented in Fig. 1 and described in more detail in Supplementary Table 1.

### Statistics and reproducibility

**Calculation of medication adherence.** First, 300 of the most prescribed active substances or active substance combinations were extracted from the database. These active substances were evaluated independently by two pharmacists who classified them as (1) active substances meant for long-term use and (2) others. The interobserver variability was assessed with intraclass correlation ($r = 0.94$, $p < 0.05$). In case of contradictions in the pharmacist's assessment, the results were validated by a medical

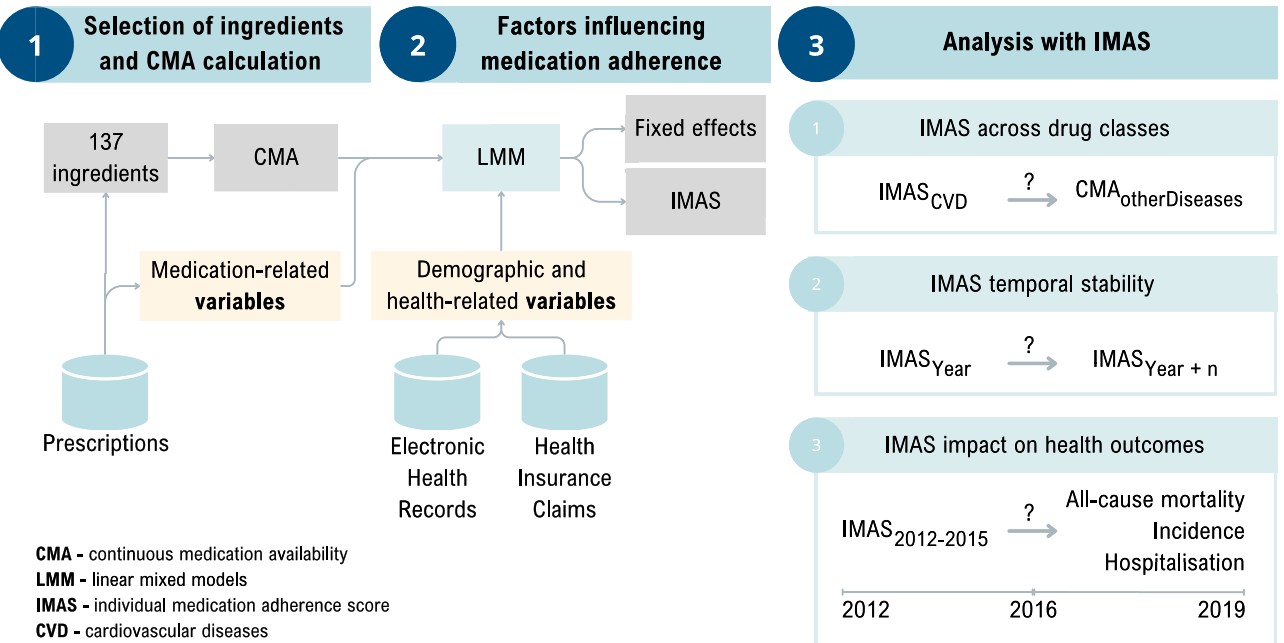

**Fig. 1 | Workflow of the analysis.** This figure shows the schematic overview of the analytical workflow. Step 1 illustrates the selection of active substances for chronic use and calculation of continuous multiple interval measures of medication availability (CMA). Step 2 shows the construction of linear mixed models (LMM) including demographic, health- and medication-related fixed effects and person as a random effect to derive the Individual Medication Adherence Score (IMAS). Step 3 presents downstream analyses, including cross-medication prediction, temporal stability assessment of IMAS, and survival analyses evaluating the association between IMAS and future health outcomes.

doctor. As a result, 137 active substances for chronic conditions were identified and included in the adherence calculations (Fig. 1, step 1). As each prescription includes diagnosis info, the most frequent diagnosis for each active substance was extracted. Based on medical expert opinion and previous studies[13], the active substances were combined into ICD-10-based diagnosis groups (Supplementary Data 1).

For medication adherence calculations, we applied continuous multiple interval measures of medication availability (CMA) on dispensed prescriptions[14]. CMA describes the proportion of days within a specified observation period during which a patient is covered with dispensed medication. CMA is calculated based on the dispensed medication and dosing regimen provided on the prescription. Out of eight CMA measures, CMA5 was used as it considers the gap times in medication availability and assumes that the new refill is banked until the previous prescription is depleted[14]. Moreover, it allows carry-over within the observation window, while any unused medication available at the end of the observation window is ignored in the calculations[14]. The CMA5 value was calculated with a yearly interval from the beginning of the first dispensing of the active substances. At least one refill within 365 days was required to be included in the analysis. The days of supply for active substances were calculated using the following information from the prescription: the amount of drugs in one package (e.g. number of tablets), the number of packages purchased and the dosage prescribed by the doctor. This allowed us to calculate the number of days for which the patient had medication available. In case of missing data, different imputation approaches were applied[15]. CMA calculations were active substance based to minimise the potential effects of medication substitutions (e.g. original vs. generic drugs) and drug scheme changes. This means that if the person received one active substance and the drug scheme was changed so that a combination of old and new active substances was provided, then the CMA5 calculations continued for the old active substance and in parallel started for the new active substance. The CMA5 was calculated with AdhereR package, which computes adherence estimates on electronic health data and visualises adherence patterns[16].

For descriptive statistics proportions, means with standard deviations, and the prevalence of good adherers (CMA ≥ 0.8) were calculated. To calculate average CMA per diagnosis group and their 95% confidence intervals (CI), we first calculated the mean CMA for each individual per active substance, then derived the individual level and disease group level average. The gender differences were analysed with the Mann–Whitney *U* test.

### Factors influencing medication adherence

To assess the effect of various demographic, health, and medication-related variables on CMA, we employed linear mixed models (LMM) using packages lme4 (v1.1-14)[17] together with package rsq (v2.6)[18] to calculate models' R-square (Fig. 1, step 2 and Supplementary Table 1). LMM was chosen for its ability to capture individual-level variability and its suitability for repeated measures. All subjects who had at least one CMA calculated for active substance under observation were included in the model as random effects, while all other covariables were treated as fixed effects. This approach allowed us to account for both within-subject changes over time and between-subject differences. For LMM, the coefficient and 95% confidence intervals are reported.

The demographical fixed effects included in the model were gender (male/ female) and age categorised into five groups: 0–19 years, 20–39 years, 40–59 years, 60–79 years, and 80 years and above. Age was treated as categorical variable to account for potential differences in effects across the age range and to enhance interpretability. BMI was included in the model as a categorical variable with four categories: (1) under and normal weight (BMI < 24.9), (2) overweight (25.0 ≤ BMI < 29.9), (3) obese (BMI ≥ 30), and (4) unknown. The occurrence of hospitalisation (yes/no), diagnosis of depression (yes/no), number of all diagnoses, and number of different active substances administered were assessed and included in the model on a yearly basis. The first occurrence of a comorbidity condition that belongs to the Charlson Comorbidity Index[19] and the first occurrence of dementia or

mental retardation diagnosis set the status for the following years. Dementia and mental retardation diagnoses were used to discriminate based on cognitive impairment, which has previously been identified as a risk factor for low medication adherence[20]. The active substance was included in the model as a categorical variable. Metoprolol was used as a reference category as this was the most frequently prescribed active substance. For each active substance, its administration route and consecutive year of active substance administration were included in the model. As each active substance was mainly prescribed for specific condition, a binary value for the most prevalent diagnosis was created. The combination of the active substance and its most prevalent diagnoses was marked as 1, while all other combinations were 0. The interaction between 137 active substances and the 12 most prevalent conditions was also included in the model. The Bonferroni correction was applied to the LMM to evaluate the statistical significance of the variables.

### Individual Medication Adherence Score (IMAS)

The Individual Medication Adherence Score (IMAS) was derived as the person-specific random effect from LMM (Fig.1, step 2 and Supplementary Table 1). This model included previously described demographic, health- and medication-related variables where person was included as random effect. Conceptually, IMAS captures an individual's inherent medication-taking behaviour or their tendency to adhere to medications—their personal "baseline" adherence tendency that persists across prescriptions and time, after accounting for demographic, medication- and health-related factors.

### Analysis with IMAS

To determine whether IMAS calculated for active substances related to one disease group helps to predict the medication adherence for other diseases, we first calculated LMM using only active substances associated with different cardiovascular diseases (CVD) (hypertension (I10-I15), arrhythmias (I46-I49), heart failure (I50), ischemic heart disease (I20-I25), other diseases of the circulatory system (I67-I70)) (CVD model) (Fig. 1, step 3 and Supplementary Table 1). Next, two LMMs were created to assess the effect on medication adherence: (1) LMM with all other active substances and, (2) LMM with all other active substances and IMAS from the CVD model (Fig.1, step 3.1 and Supplementary Table 1). Similarly to previous models demographical, health- and medication-related variables were added as fixed effects in all these models. This approach allowed us to evaluate how medication adherence calculated for some active substances affects the medication adherence of other active substances.

To assess the stability of the IMAS over time, we calculated IMAS using LMM with demographical, health- and medication-related variables as fixed effects for each year (Fig.1, step 3.2 and Supplementary Table 1). Next, the correlation between yearly IMAS was calculated.

For survival analysis, we created LMM for medication adherence with demographical, health- and medication-related variables as fixed effects and person as random effect using data from 2012 to 2016 (Fig.1, step 3.3 and Supplementary Table 1). From this model IMAS was extracted and included in the Cox proportional hazards model to estimate incidence and hospitalisation for selected diseases (Supplementary Table 2) between 2017 and 2019. Only persons who were alive on 01.01.2017 and did not have an occurrence of chronic condition under observation or hospitalisation from 2012 to 2016 were included in the analysis. The diagnoses included in the analysis were based on previous studies[13,21] and medical expert opinion. In the Cox proportional hazards model, the IMAS was continuous variable, and the model was controlled for year of birth and gender. The Bonferroni correction was applied to evaluate the significance of hazard ratios. The hazard ratios, together with 95% CI, are presented. Significant IMAS was visualised with Kaplan–Meier curve. All analyses were performed using R (v4.3.3).

The study was approved by the Research Ethics Committee of the University of Tartu (300/T-23) and the Estonian Committee on Bioethics and Human Research (1.1-12/653). The study used existing pseudonymized

health data, and in accordance with national data protection legislation, the requirement for informed consent was waived.

## Results

### Medication adherence

Less than half of the subjects in the database (43.0%, $N = 64,837$) had dispensed at least one active substance under observation twice within a year, and it was possible to calculate CMA. That group of patients included 57% females with an average age of 56.5 ± 21.8 years (Table 1).

On average, 75% of the days were covered with dispensed active substance (Table 1). This means that a subject was in possession of active substance for 274 days within a year. There were 52.1% of subjects with an average CMA over 0.8 and 7.7% with an average CMA below 0.4. This corresponds to possession of active substance for 292 and 146 days per year, respectively. Overall, females had significantly higher average CMA compared with males (0.76, 95% CI 0.76–0.77 vs 0.73, 95% CI 0.72–0.74,

respectively, $p < 0.001$). At least one comorbidity from the Charlson Comorbidity Index was present for 64.9%, and depression was diagnosed for 21.7% of subjects.

When looking at the raw CMA values, the subjects took on average 4.3 ± 3.7 different active substances during the study period. The CMA per active substance ranged from 0.423 (albuterol, 95% CI 0.414–0.432, corresponds to 154 days per year) to 0.922 (warfarin, 95% CI 0.917–0.926, corresponds to 337 days per year) (Supplementary Data 2). The active substances were assigned to a disease groups based on the most prevalent diagnoses on their prescription (Supplementary Data 1). Eight disease groups had an average CMA of 0.8 or higher which means that at least 292 days per year were covered with medication—disorders of thyroid gland (E00–E07), presence of cardiac and vascular implants and grafts (Z95), malignant neoplasm of breast (C50), Parkinson disease (G20), arrhythmias (I46–I49), malignant neoplasm of prostate (C61), osteoporosis (M80, M81), and Glaucoma (H40-H42) (Table 2). The lowest medication adherence was for diseases of the digestive system (K00–K93) (CMA = 0.488, 95% CI 0.479–0.497). The highest proportion of adherent subjects (CMA ≥ 0.8) was for malignant neoplasm of breast (82.0%) and disorders of thyroid gland (78.4%), while the lowest proportion of adherent subjects was among diseases of respiratory system (22.7%) and digestive system (24.3%) (Table 2).

Methylphenidate users were the youngest (13.6 years, 95% CI 12.9–14.3), while donepezil and isosorbide users were the oldest (77.4 years, 95% CI 76.5–78.3 and 77.3 years, 95% CI 76.9–77.7, respectively) (Supplementary Data 3).

### Factors influencing medication adherence

The CMA variation was modelled using linear mixed models (LMM), including several demographic, health- and medication-related variables as fixed effects (Supplementary Table 1). Adding person as a random-level effect allowed to capture the correlation between the adherence values across medications and time periods for the same person and highlight the person-

## Table. 1 | Descriptives of 64,837 subjects included in the analysis

| Characteristics | $N$ = 64,837 |
|---|---|
| Gender, $N$ (%) | |
| Female | 37,111 (57.2%) |
| Male | 27,726 (42.8%) |
| Body mass index category, $N$ (%) | |
| Obese | 10,690 (16.5%) |
| Overweight | 9396 (14.5%) |
| Under or normal weight | 8003 (12.3%) |
| Unknown | 36,748 (56.7%) |
| Mean age in years (sd) | 56.50 (21.75) |
| Mean CMA (sd) | 0.75 (0.21) |

## Table. 2 | Average CMA with 95% confidence intervals and proportion of adherent persons (CMA ≥ 0.8) by disease groups

| Disease (ICD-10 code) | Average CMA | 95% CI | Adherent (%) |
|---|---|---|---|
| Disorders of thyroid gland (E00-E07) | 0.871 | 0.867…0.875 | 78.4 |
| Presence of cardiac and vascular implants and grafts (Z95) | 0.869 | 0.863…0.875 | 76.3 |
| Malignant neoplasm of breast (C50) | 0.866 | 0.852…0.880 | 82.0 |
| Parkinson disease (G20) | 0.829 | 0.818…0.839 | 67.9 |
| Arrhythmias (I46-I49) | 0.826 | 0.822…0.830 | 66.2 |
| Malignant neoplasm of prostate (C61) | 0.820 | 0.803…0.838 | 65.8 |
| Osteoporosis (M80, M81) | 0.812 | 0.801…0.823 | 66.7 |
| Glaucoma (H40-H42) | 0.802 | 0.798…0.805 | 58.0 |
| Rheumatoid arthritis and related disorders (M05, M06, M08, M13, M30-35, M45) | 0.797 | 0.789…0.805 | 59.9 |
| Diabetes (E10-E14) | 0.792 | 0.790…0.794 | 58.2 |
| Hypertension (I10-I15) | 0.791 | 0.789…0.792 | 57.4 |
| Disorders of lipoprotein metabolism and other lipidaemias (E78) | 0.775 | 0.772…0.778 | 55.3 |
| Epilepsy (G40-G42) | 0.773 | 0.764…0.782 | 56.1 |
| Diseases of male genital organs (N40-N51) | 0.771 | 0.766…0.776 | 55.0 |
| Psychotic, mood and neurotic disorders (F20-F48, F90) | 0.764 | 0.761…0.767 | 53.1 |
| Dementia (F00-F03, G30, G31) | 0.747 | 0.733…0.761 | 51.8 |
| Gout (M10) | 0.730 | 0.723…0.736 | 46.2 |
| Heart failure (I50) | 0.694 | 0.690…0.699 | 38.8 |
| Ischemic heart disease (I20-I25) | 0.676 | 0.669…0.683 | 39.4 |
| Other diseases of the circulatory system (I67-I70) | 0.668 | 0.659…0.677 | 39.1 |
| Chronic diseases of respiratory system (J43-J47) | 0.554 | 0.549…0.560 | 22.7 |
| Diseases of the digestive system (K00-K93) | 0.488 | 0.479…0.497 | 24.3 |

specific effect in medication adherence value. The whole model described 33.6% of the variation in medication adherence measurements, while 22.0% of that was explained by the person-specific effects or in other words by IMAS. From demographic variables, gender had no effect on medication adherence, while 80 years and older people had higher (0.11, 95% CI 0.10–0.13) medication adherence compared to those younger than 20 years (Fig. 2, Supplementary Data 4). Several health-related factors like the occurrence of depression, dementia, hospitalisation or comorbidities did not have an effect on medication adherence. The expected CMA values were more than 0.1 units lower for 27 and higher for two medications compared to metoprolol, which was used as the comparator in the analysis. Some diagnoses were associated with adherence. In most cases, the effect was less than 0.1, which corresponds to 37 days per year. However, very few like malignant neoplasm of breast (C50) (0.30, 95% CI 0.26–0.34) and glaucoma (H40) (0.35, 95% CI 0.21–0.49) displayed comparably large effect sizes to medication adherence. In other words, when patients are otherwise comparable in terms of covariates, those with malignant neoplasm of the breast or glaucoma exhibit 110–128 more days of medication supply.

### Individual Medication Adherence Score (IMAS)

IMAS is the person-specific effect extracted from LMM, including several demographic, health- and medication-related variables. It represents an individual's inherent medication-taking behaviour—their personal "baseline" adherence tendency that persists across prescriptions and time, after accounting for demographic, medication—and health-related factors. IMAS ranges from −0.51 to 0.50 (Fig. 3). This means that there are subjects who demonstrate approximately half a year less medication supply (IMAS = −0.51) than other, otherwise similar individuals.

### Analysis with IMAS

IMAS was included in further analysis to evaluate its impact and continuity over time. First, the IMAS calculated for only active substances prescribed for cardio-vascular diseases (CVD) was added into a LMM calculated for all other active substances prescribed for chronic conditions (non-CVD). The inclusion of IMAS based on CVD active substances significantly increased the medication adherence for non-CVD active substances (estimate = 0.45, $p < 0.001$). Moreover, the R-squared of fixed effects was slightly higher for a model where IMAS from the CVD model was added as fixed effects compared to the model without this IMAS ($R^2 = 0.14$ vs $R^2 = 0.11$, respectively).

Next, the continuity of the IMAS over time was assessed. The correlation between yearly IMAS declined over time, with the highest correlation being between consecutive years ($r_{avg}$ = 0.46, range 0.36–0.49, $p < 0.05$) (Fig. 4).

Lastly, we sought to evaluate if IMAS predicts health outcomes in the future. For this purpose, IMAS calculated for the period 2012 to 2016 was used to predict the health outcomes in 2017–2019 using a Cox proportional hazards model. From 2017 to 2019, there were 5904 hospitalisations. We found that higher IMAS reduced the risk of overall hospitalisation (HR = 0.76, 95% CI 0.60–0.97, $p < 0.05$), but no effect on cause-specific hospitalisation risk after Bonferroni adjustment was detected (Supplementary Table 3).

We used the same approach to predict the incidence of a collection of chronic conditions. Out of 32 chronic conditions, higher IMAS was associated with a lower risk of cause-specific incidence for 17 conditions after Bonferroni correction. The reduction in incidence risk ranged from 27% for atherosclerosis/PAOD and diseases of liver to 58% for diseases of stomach (Fig. 5 and Supplementary Table 4). In other words, better individual medication-taking behaviour indicates a protective effect against adverse health outcomes.

## Discussion

This study set out to investigate medication adherence and its impact on health outcomes by analysing more than 137 active substances alongside a comprehensive set of demographic, health- and medication-related variables from real-world electronic health data. We modelled adherence values using LMM, revealing low predictive power for many commonly studied risk factors, but consistent person-level behaviour across medications and time. The calculated IMAS was predictive for both future adherence as well as health outcomes.

Our extensive analysis across 137 active substances revealed a significant variation in medication adherence based on active substance and disease—there was more than a two-fold difference between active substances with the best and the worst adherence, and almost a four-fold difference in the prevalence of good adherence between diseases. The prevalence of patients with good adherence in our study was slightly lower for antidiabetic agents (58%) and similar for statins (55%) and antihypertensives (57%) compared with the previous meta-analysis[22]. However, the diversity in adherence calculation methods and active substance selection complicates the comparison between studies and diseases. Also, we expect differences in adherence patterns between localities and healthcare systems to be present, raising the need to conduct similar studies with consistent methodology on the growing number of population-based prescription databases worldwide.

It is widely accepted that medication adherence is influenced by several individual, medication-related and healthcare system-related factors[3,7,23]. Our results indicated that a considerable amount of observed variables had a statistically significant impact on medication adherence, but the effect size was very small. This finding is in accordance with previous studies where the observed effects were very small and often inconsistent between studies for most patient-related factors[7,8]. More importantly, we found that almost a quarter of the medication adherence variance was explained by IMAS, capturing more variation than the fixed effects in the mixed-effect model. The significance of IMAS has several implications for further medication adherence research.

First, the medication-taking behaviour represents a broader patient-level phenomenon that manifests consistently across therapeutic classes, and it should be studied as such. In current study, the individual medication-taking behaviour deviated by approximately six months above or below the mean level of adherence. There is a need for a better understanding of medication adherence behaviour through larger studies on more active substances in diverse patient cohorts across the world. Also, when obtaining more precise individual adherence estimates, we can start characterising the determinants by integrating prescription datasets with rich individual-level data such as socioeconomic status, personality characteristics, and genetic profiles. The substantial year-to-year variability in adherence patterns in the current study indicates that medication-taking behaviour is shaped by both enduring patient characteristics and dynamic life circumstances. In future research, it is important to tease these effects apart.

Second, the strong effect of IMAS in our study points to the need for more personalised adherence interventions rather than population-level strategies. Our method using IMAS seems a promising approach to identify high-risk patients and to provide timely interventions, especially as IMAS had better predictive power compared to different demographic, health- and medication-related factors, even across medication classes. Therefore, we believe that IMAS can significantly contribute to more effective personalised risk models which take into account medication administration history across different health conditions and medications.

Generally, better medication adherence has been associated with a lower risk of several diagnoses[24,25] and hospitalisation[4,24,26]. We showed a 24% reduction in the risk of overall hospitalisation and a 27–58% reduction in the incidence risk for several diseases among those with higher IMAS. However, the decrease in the incidence risk was not observed for all diseases. This highlights the nuanced relationship between adherence and health outcomes. Proper administration of necessary medications certainly helps to manage chronic disease, but patients with better adherence may visit their healthcare providers more frequently, resulting in earlier and potentially more diagnoses. At the same time, it could be hypothesized that a significant deterioration of health or an acute episode could have a positive effect on medication adherence. Decomposing such a complex causal network is an important next step in understanding the phenomenon. However, it requires a different set of methodologies and is out of the scope of the current paper.

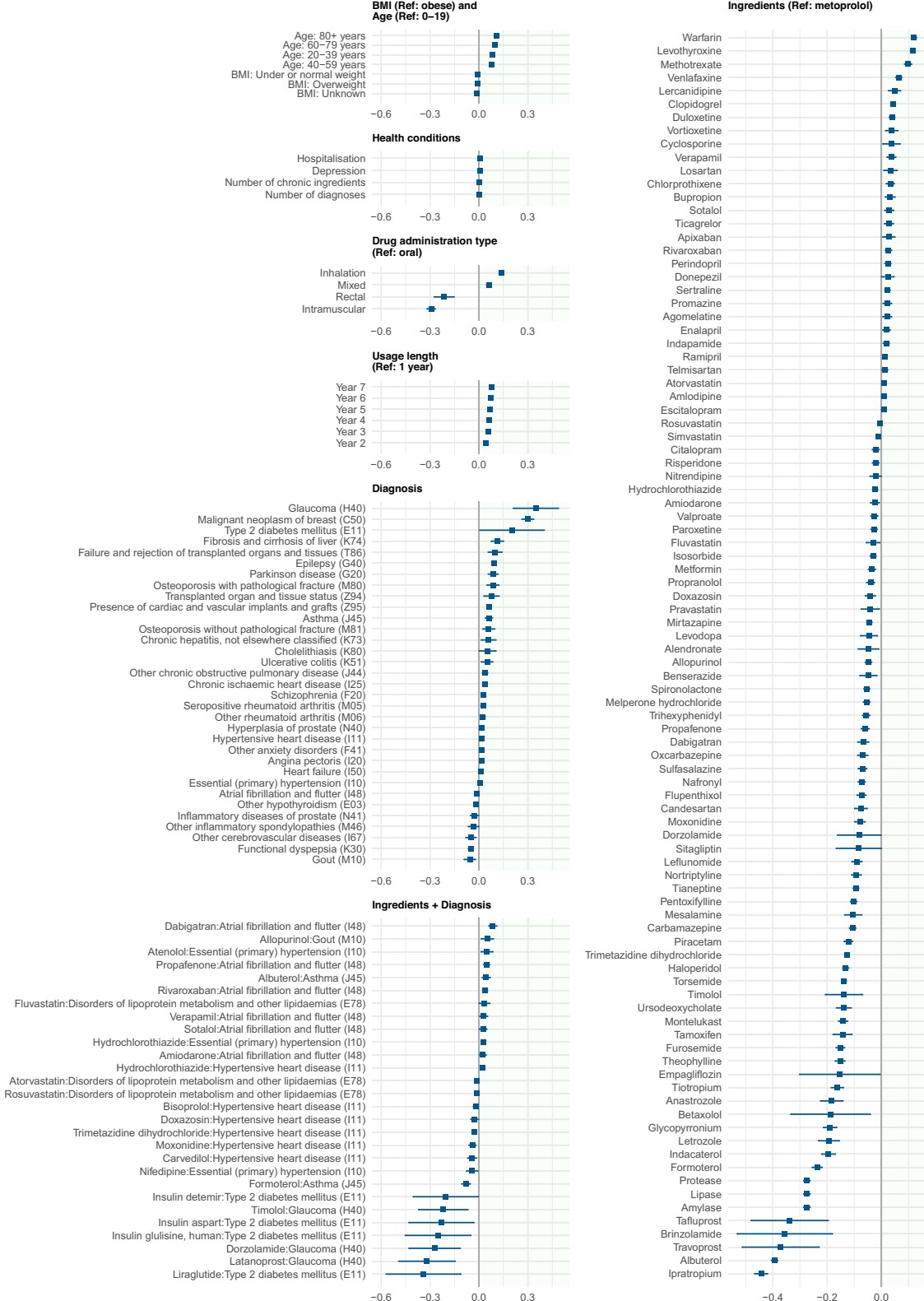

**Fig. 2 | Variables influencing medication adherence based on linear mixed model analysis.** Effect estimates and 95% confidence intervals from the linear mixed model assessing demographic, health- and medication-related factors associated with medication adherence (CMA), including $n = 64\,837$ subjects. The model includes person as a random effect to account for repeated measurements within individuals.

Only variables that remain statistically significant after Bonferroni correction are shown. Effect sizes represent the estimated change in CMA associated with each variable compared to its reference category. Information for all included variables is presented in Supplementary Table 4.

The study has limitations that need to be acknowledged. When interpreting the results, it should be kept in mind that a large proportion of BMI values were missing. We assume missing at random, however, if this is not the case, our estimates could be biased. One limitation of using prescription data is the lack of information of whether or how the medication was actually taken. Also, any dosing changes by doctors after the medication has already been dispensed remain undetected. A crucial aspect of calculating adherence measures on prescription data is the accuracy and completeness of the records. In some countries, the completeness of prescription data is hindered by the diversity of insurers or pharmacies with independent, non-linkable systems. However, that is not the case in our study, as the national prescription database stores information about all issued and dispensed prescription medications. Still, the day's supply was originally missing for 38% of the prescriptions. The missing information was replaced with imputation, resulting in day's supply value for almost all prescriptions. The imputation process and its effect on the CMA is described in more detail elsewhere[15]. Although the imputation may potentially cause a systematic bias, it is taken into account in medication-based fixed effects, and thus, its influence on IMAS can be considered modest.

At the same time, the strengths of our study are analysing more than 137 active substances in a consistent manner in over 60 thousand patients, allowing us to detect more subtle effects and perform accurate between-active substance comparisons. Moreover, our analysis takes into account a wide range of factors from medical records potentially influencing medication adherence, thus, reducing apparent confounding effects. The study provides a framework for further medication adherence research on real-world health databases modelling person-level variation while taking into account linked health events and other data modalities. Further aspects of medication adherence, such as primary adherence and persistence, should also be included in building more accurate prediction methods for identifying high-risk patients.

Our study highlights the value and possibility of integrating large-scale real-world data for proactive identification of at-risk patients. The study warrants a shift from specific medication and population-based focus towards a more patient-level perspective in adherence research. The predictiveness of the IMAS suggests the potential for personalised interventions in clinical practice and more efficient public health strategies and policies.

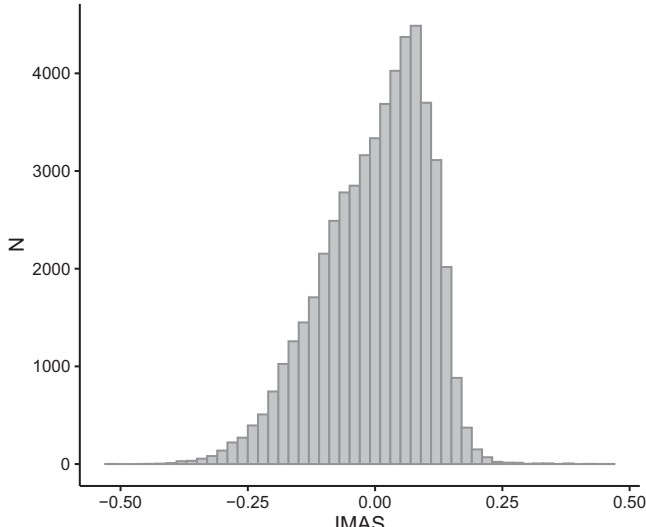

**Fig. 3 | Distribution of Individual Medication Adherence Score (IMAS).** Distribution of the Individual Medication Adherence Score (IMAS), derived as the person-specific random effect from the linear mixed model adjusting for demographic, health- and medication-related variables. IMAS represents the deviation from the population-average adherence after accounting for observed covariates. Positive values indicate higher-than-average baseline adherence tendency, whereas negative values indicate lower-than-average adherence tendency.

**Fig. 4 | Two-sided correlations of yearly IMAS estimates from 2012 to 2019.** Pairwise correlations between yearly IMAS estimates calculated separately for each calendar year using linear mixed models with identical fixed effects structure. Correlation coefficients quantify the temporal stability of person-level adherence tendencies across years. All displayed correlations are statistically significant after Bonferroni adjustment ($p < 0.001$).

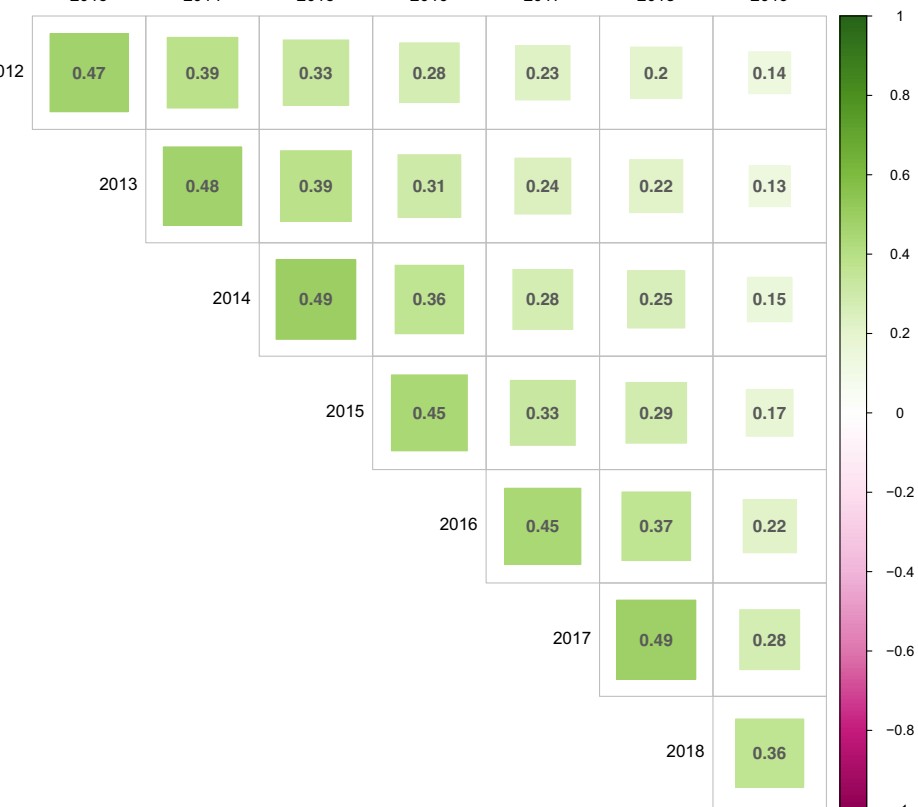

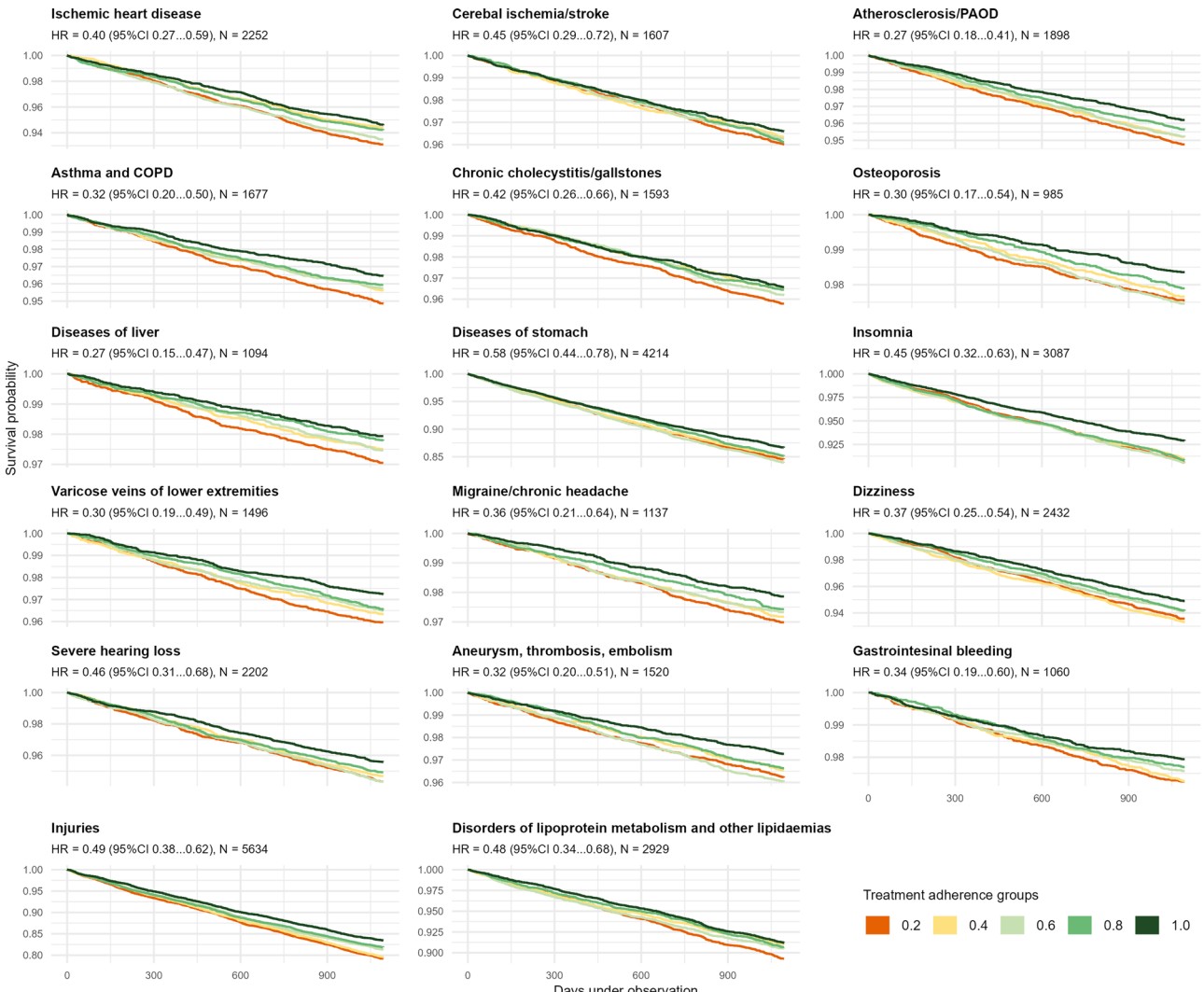

**Fig. 5 | Kaplan–Meier curves for cause-specific incidence for statistically significant diagnoses.** Kaplan–Meier curves illustrating time-to-event analyses for selected chronic conditions significantly associated with IMAS in Cox proportional hazards models. IMAS is derived from earlier observation years and included as a predictor of incident disease during follow-up. Curves demonstrate differences in cumulative incidence according to IMAS levels. Models are adjusted for year of birth and gender, and statistical significance is evaluated with Bonferroni correction.

## Data availability
The datasets generated and analysed during the current study are not publicly available due to legal restrictions on sharing de-identified data. According to legislative regulation and data protection law in Estonia, the authors cannot publicly release the data received from the health data registries in Estonia. However, the data can be requested by completing necessary applications in order to carry out research or an evaluation of public interest and acquiring the permission of the controller of the databases (https://www.tehik.ee/en/statistics). The timeline for review and approval depends on the scope of the request and is determined by the data controller. However, it typically takes approximately 4–8 weeks. More information about data availability: Kerli.Mooses@ut.ee. The source data for Fig. 2 are in Supplementary Data 4.

## Code availability
The underlying code for this study cannot be made publicly available at this time because it contains components linked to secure data infrastructure. A simplified version of the analytical scripts can be made available from the corresponding author upon reasonable request.

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

## Acknowledgements

This study was co-funded by the European Union and Estonian Ministry of Education and Research via project TEM-TA72 and Estonian Research Council grants PRG1844 and PSG809. Also, this study has received funding from the European Union's Horizon Europe research and innovation programme under grant agreement No 101060011. Views and opinions expressed are however, those of the author(s) only and do not necessarily reflect those of the European Union or European Research Executive Agency. Neither the European Union nor the granting authority can be held responsible for them. This research was co-funded by the European Union through the European Regional Development Fund (Project No. 2021-2027.1.01.24-0444) and Estonian Ministry of Education and Research (Teaming for Excellence).

## Author contributions

K.M., R.K., T.T., H.L., N.U., S.R., J.V., J.H., and M.O. participated in the design of the study and methodology. The literature search was carried out by K.M. and H.K. Data curators were S.T., M.O., and M.M. Software developments were done by J.H., M.O., and M.M. The analysis was conducted by K.M., H.L., and R.K. Figures were designed by MP. Funding acquisition—R.K., S.R., and J.V. The original draft was written by K.M. while all authors read and edited the manuscript and approved the final version.

## Competing interests

The authors declare no competing interests.
