## [Transparent Peer Review file · Communications Medicine]

Systematic evaluation of medication adherence determinants across 137 active substances on population-level real-world health data

Corresponding Author: Dr Kerli Mooses

Version 0:

Reviewer comments:

Reviewer #1

(Remarks to the Author)

Novelty:

This paper highlights important phenomena that is often missed in many adherence studies and generally not considered adequately by developers of adherence interventions. Namely that the majority of variation in medication adherence behaviour is not explained by known factors, that both patient level and non-patient level factors contribute to adherence behaviours, of these two patient level factors explain more variation and patient level factors influence on adherence behaviour are not temporally stable. This is important in light of many historical failures of simple adherence interventions, non-personalised adherence interventions, and one off interventions. This research hopefully serves to guide more appropriate development of these public health initiatives.

Overall my major concerns are with lack of clarity on the study design to properly evaluate the potential biases and limitations. Specifically it is difficult to disentangle the study methodology from the statistical analysis, and I would rate this as having a high risk of bias for many potential pharmaco-epidemiological cardinal sins. While the graph outlining calculation of IMAS is helpful, a graph outlining study design would be more useful (please refer to Schneeweiss 2019, Graphical Depiction of Longitudinal Study Designs in Health Care Databases)

Methodological questions to help clarify:

1. The study design, patient inclusion and exclusion criteria, and index date are unclear (major)
2. Relatedly, it is unclear how and when the IMAS is defined for a patients, and then how it is applied in the analysis (major)
3. Why do you treat age as a nominal (unordered) categorical variable in your modelling (minor)
4. Please discuss limitations of large proportion of missing values for BMI, missing at random assumptions and impact it may have on modelling (major)
5. Please discuss limitations of large proportion of missing values for days supply of medication. It may need more justification and explanation (major)

Reviewer #2

(Remarks to the Author)

Please address the following comments:

1. Please describe the methods section in detail. This section should include the inclusion and exclusion criteria, covariates of interest, analysis plan, and any other relevant information that will help researchers replicate this study. The methods section should be after the introduction section
2. Explain how the average CMA was calculated for the disease mentioned in Table 2 (this should be included in the analyses section).

3. How did the authors identify the 137 ingredients? For atrial fibrillation, dabigatran and rivaroxaban were selected as the treatment. Why were other direct oral anticoagulants not selected
4. "The CMA variation was modeled using linear mixed models (LMM), including several demographic, health- and medication-related variables as fixed effects, as well as random person-level effects, capturing the correlation between the observed adherence values across medications and periods for the same person." Explain the statement for clarity.
5. "However, very few like malignant neoplasm of breast (C50) (0.30, 95% CI 0.26–0.34) and glaucoma (H40) (0.35, 95% CI 0.21–0.49) displayed comparably large effect sizes to medication adherence". So what does this imply? Please include it as well.
6. Are there any other studies that used the same approach? The method taken by the authors does not convince me. How can we justify comparing metoprolol to 136 other medications? The 136 medications can be used for different indications, have varying doses, supply values, quantities, costs, and side effects.
7. How was IMAS calculated? Explain.
8. What variables were included in the survival analyses?
9. "Out of 32 chronic conditions, higher IMAS was associated with a lower risk of cause-specific incidence for 17 conditions after Bonferroni correction". Please clarify if this means higher IMAS to 137 medications? Or does it include only medication for a specific disease? I am not sure how warfarin can reduce the incidence of osteoporosis. There is no causal relation or association between warfarin and osteoporosis.
10. Please remove Figure 3. Most of the drugs do not have any impact or association on the outcomes given in Figure 3. Earlier studies have reported that the association between adherence and reduced injuries is biased.

Reviewer #3

(Remarks to the Author)

The study uses a robust methodology, consistent selection of chronic-use active pharmaceutical ingredients, and advanced statistical modeling. The concept that medication-taking behavior is a stable, individual-level trait that transcends therapeutic classes is novel and impactful. The idea of using IMAS as a predictor for adherence and health outcomes may represent a step forward in personalized medicine.

The results are of immediate relevance to both pharmacy and broader health research. For pharmacists, it offers a tool (IMAS) for identifying patients at risk of nonadherence across multiple medications; For policymakers, it demonstrates how real-world prescription data can support risk stratification and targeted interventions; and for Public health impact, it proposes a scalable, data-driven framework to understand adherence in the population.

Some comments:

1. Throughout the manuscript, the authors use the term "ingredient" to refer to the pharmacologically active component of a medicinal product. However, the correct and internationally accepted terminology in pharmaceutical sciences, particularly under the European Medicines Agency (EMA) and International Council for Harmonisation (ICH) guidelines, is "active substance" or "active pharmaceutical ingredient (API)." Most peer-reviewed publications on medication adherence and pharmacoepidemiology use "active substance" or "API." Consistency with this terminology facilitates better indexing, searchability, and comprehension across disciplines.
2. I recommend that the authors briefly explain what CMA5 entails and what the AdhereR package does, ideally with one or two sentences summarising its key assumptions. This will improve accessibility of the methods section for interdisciplinary readers, particularly clinicians, pharmacists, and public health professionals who may not be familiar with adherence algorithms implemented in R.
3. In addition, please explicitly state whether CMA5 was calculated using dispensing data only, or if prescription issuance was also factored in. Based on current phrasing, it appears that dispensed prescriptions were used, but this should be clearly confirmed. In some parts of the manuscript (e.g., methods section), the text refers to "prescribed medications," while in others it mentions "purchased prescriptions" or the use of pharmacy claims data. This could create confusion for readers regarding the true basis of the adherence calculations.
4. The concept of IMAS is central to the manuscript and one of its most innovative contributions. However, its definition, purpose, and interpretability are not clearly explained for a broader audience. I recommend: Introducing IMAS more clearly and accessibly in the Introduction, with a concise definition in plain language; Including a brief conceptual summary in the Results section alongside the statistical explanation; Adding a sentence to the Abstract highlighting IMAS as a novel person-level metric that predicts both adherence and health outcomes.

I also recommend to address the following additional comments:

5. 38% of prescriptions lacked information on days of supply and were imputed. Though addressed via fixed effects and ingredient-level modeling, this could introduce medication-specific biases?
6. Adherence was estimated from refill behavior (? see above), not actual consumption. This is a common limitation, but it should be highlighted more clearly, at least in the discussion.
7. While IMAS shows year-over-year stability, the potential confounding effects of acute health events or life changes on adherence trajectories are mentioned but not modeled directly?

This is a high-quality, original contribution to the literature on medication adherence and real-world health data analytics. Minor refinements would strengthen an already impactful manuscript.

Reviewer comments:

Reviewer #2

(Remarks to the Author)

Dear Authors,

Thank you for addressing all the comments. The major revision has significantly improved the quality of the manuscript, and I agree to its publication.

Reviewer #3

(Remarks to the Author)

The revision improves clarity, methods transparency, and the centrality/interpretability of IMAS. The paper now consistently uses active substance/API terminology, clearly explains CMA5/AdhereR, and distinguishes dispensed (purchased) data from prescribing. The new workflow figure and expanded Methods/Supplement clarify the study design and analysis steps. I recommend acceptance.

We thank all the reviewers for their valuable and constructive comments which have helped to improve the clarity and quality of our manuscript.

Reviewer #1 (Remarks to the Author):

Novelty:

This paper highlights important phenomena that is often missed in many adherence studies and generally not considered adequately by developers of adherence interventions. Namely that the majority of variation in medication adherence behaviour is not explained by known factors, that both patient level and non-patient level factors contribute to adherence behaviours, of these two patient level factors explain more variation and patient level factors influence on adherence behaviour are not temporally stable. This is important in light of many historical failures of simple adherence interventions, non-personalised adherence interventions, and one off interventions. This research hopefully serves to guide more appropriate development of these public health initiatives.

Overall my major concerns are with lack of clarity on the study design to properly evaluate the potential biases and limitations. Specifically it is to difficult to disentangle the study methodology from the statistical analysis, and I would rate this as having a high risk of bias for many potential pharmaco-epidemiological cardinal sins. While the graph outlining calculation of IMAS is helpful, a graph outlining study design would be more useful (please refer to Schneeweiss 2019, Graphical Depiction of Longitudinal Study Designs in Health Care Databases)

Methodological questions to help clarify:

1. The study design, patient inclusion and exclusion criteria, and index date are unclear (major).

Thank you for pointing out this shortcoming. Additional information about study design, time periods and covariates has been included to the Methods section. Moreover, to give a better overview of study design and covariates included into different analysis, we added a Supplementary table 1.

2. Relatedly, it is unclear how and when the IMAS is defined for a patients, and then how it is applied in the analysis (major)

Hopefully the Supplementary table 1 helps to clarify when and how the IMAS is defined and applied.

3. Why do you treat age as a nominal (unordered) categorical variable in your modelling (minor)

Thank you for this valuable comment. There were two main reasons why we used age as categorical variable. First, we cannot assume that the age effect is similar across the age range, therefore, using age groups can help to better identify such effects. Secondly, we categorized age in our analysis to enhance interpretability. Age groups (e.g., children and adolescents (0-19 years), young adults (20-39 years), middle-aged adults (40-59 years), older adults (60-79 years), elderly (80+ years) are more intuitive and align with how findings are often communicated and applied in clinical and public health contexts. Presenting age in categories allows readers, practitioners, and policymakers to more readily understand and compare risk across meaningful subpopulations. For these reasons, we considered categorical modelling more suitable for our study objectives. We have added a short reasoning to the manuscript Methods section.

4. Please discuss limitations of large proportion of missing values for BMI, missing at random assumptions and impact it may have on modelling (major)

We have highlighted this limitation in revised manuscript and added a following sentences: “When interpreting the results, it should be kept in mind that a large proportion of BMI values were missing. We assume missing at random, however, if this is not the case, our estimates could be biased. “

5. Please discuss limitations of large proportion of missing values for days supply of medication. It may need more justification and explanation (major)

Originally 38% of day's supply values were missing. However, after applying the imputation on daily dose values we were able to calculate days' supply to all prescriptions. During the review period, we have published a preprint where the imputation and its effect is described in more detail (Malk et al, 2025). This analysis also showed that the missing information was not active substance specific. Most of the missing values of day's supply originated from 2012–2016. These were the first years of implementing the digital prescription system in Estonia, and during this time daily dose information was not compulsory for the doctors. After improvement of the system when more input fields were made compulsory, the proportion of missing values reduced significantly. However, further studies with more databases can help us to detect possible imputation biases.

We have modified the wording concerning the limitations of day's supply values to better explain applied approach and its possible effect on the outcome, and we have added the reference to a preprint where the imputation is described in detail.

Reviewer #2 (Remarks to the Author):

Please address the following comments:

1. Please describe the methods section in detail. This section should include the inclusion and exclusion criteria, covariates of interest, analysis plan, and any other relevant information that will help researchers replicate this study. The methods section should be after the introduction section.

To address this comment, we have updated the Methods section. Moreover, to give a better overview of covariates, time periods, active substances included into different models a Supplementary table 1 has been added. The methods section has been moved after the introduction section.

2. Explain how the average CMA was calculated for the disease mentioned in Table 2 (this should be included in the analyses section).

Thank you for pointing out this shortcoming. We have amended the Methods section and added the information about how the average CMA was calculated for Table 2.

3. How did the authors identify the 137 ingredients? For atrial fibrillation, dabigatran and rivaroxaban were selected as the treatment. Why were other direct oral anticoagulants not selected

The selection of active substances that were included into the study was made from 300 of the most prescribed active substances or active substance combinations in our database. Out of those 300 active substances two pharmacists independently selected those that were meant for chronic diseases. Other direct anticoagulants were not that prevalent during the study period and thus, were left out from the analysis. It also should be kept in mind that we used data from 2012-2019, so newer active substances were not present or were very rarely prescribed. Moreover, over-the-counter medications (such as aspirin) were not included into the analysis. We are working towards getting an access to a more recent data, which would allow to broaden the range of active substances included into the study.

4. "The CMA variation was modeled using linear mixed models (LMM), including several demographic, health- and medication-related variables as fixed effects, as well as random person-

level effects, capturing the correlation between the observed adherence values across medications and periods for the same person." Explain the statement for clarity.

We have rephrased this sentence and hopefully made it clearer to the reader.

"The CMA variation was modelled using linear mixed models (LMM), including several demographic, health- and medication-related variables as fixed effects. Adding person as a random-level effect, allowed to capture the correlation between the adherence values across medications and time periods for the same person and highlight the person-specific effect in medication adherence value."

5. "However, very few like malignant neoplasm of breast (C50) (0.30, 95% CI 0.26–0.34) and glaucoma (H40) (0.35, 95% CI 0.21–0.49) displayed comparably large effect sizes to medication adherence". So what does this imply? Please include it as well.

We have added an interpretation of the finding to the Results section as recommended:

"However, very few like malignant neoplasm of breast (C50) (0.30, 95% CI 0.26–0.34) and glaucoma (H40) (0.35, 95% CI 0.21–0.49) displayed comparably large effect sizes to medication adherence. In other words, when patients are otherwise comparable in terms of covariates, those with malignant neoplasm of the breast or glaucoma exhibit 110–128 more days of medication supply."

6. Are there any other studies that used the same approach? The method taken by the authors does not convince me. How can we justify comparing metoprolol to 136 other medications? The 136 medications can be used for different indications, have varying doses, supply values, quantities, costs, and side effects.

To the best of our knowledge our study is the first to apply linear mixed models with person as a random-effect and across such a wide range of active substances. The rationale of our approach originates from the idea that although the active substances included into the study are prescribed for distinct diagnosis and for medical expert they can be incomparable, we could argue that from the patient's perspective each medication constitutes simply another drug prescribed by the physician. Therefore, if the patient takes one medication correctly, it may indicate how well he/she would adhere to another medication prescribed for another disease. Thus, our approach focuses on describing the patient-specific behaviour that is persistent regardless of the active substance, its dose, treated disease etc. Adding a wide range of active substances into the model gives a more comprehensive overview of the actual adherence patterns.

In the future we plan to complement current model with additional covariates, such as cost of the drug, amount of co-payment, and personality traits, which all can affect the medication adherence.

7. How was IMAS calculated? Explain.

IMAS represents the person-specific effect that is extracted from the LMM. In the LMM we add all the covariables (see Supplementary table 1) as fixed effects and person as random effect. In our model this random effect describes the person-specific medication adherence which we have named as individual medication adherence score (IMAS).

To more clearly describe the calculation of IMAS and its representation, we have modified the paragraph about IMAS and its calculation in the Methods section, added Supplementary Table 1 and a separate subsection to Results, where IMAS values are described in more detail.

8. What variables were included in the survival analyses?

For survival analysis, we created LMM for medication adherence with demographical, health- and medication-related variables as fixed effects and person as random effect using data from 2012 to 2016 (see Supplementary table 1). From this model IMAS was extracted and included in the Cox proportional hazards model which was controlled for year of birth and gender.

We hope that the amendments in the Method section and adding the Supplementary table 1 help to clarify this.

9. "Out of 32 chronic conditions, higher IMAS was associated with a lower risk of cause-specific incidence for 17 conditions after Bonferroni correction". Please clarify if this means higher IMAS to 137 medications? Or does it include only medication for a specific disease? I am not sure how warfarin can reduce the incidence of osteoporosis. There is no causal relation or association between warfarin and osteoporosis.

The IMAS was calculated across 137 active substances, controlled for fixed effects in the model (see Supplementary table 1). This IMAS was added to the survival analysis. Conceptually IMAS describes persons' tendency to adhere to medications. Based on the history of medication adherence across all administered active substances under observation, person-specific effect is calculated. Our analysis confirmed that IMAS calculated on selected active substances predicted the medication adherence of other active substances. Moreover, in the added subchapter in the Results section we show that IMAS ranges from -0.51 to 0.50. This means that the individual medication-taking behaviour deviates by approximately six months above or below the mean level of adherence. Taken this all into account, this medication-taking behaviour in turn has an effect on the incidence of some diseases.

We hope the amendments in the manuscript have clarified this idea to the readers.

10. Please remove Figure 3. Most of the drugs do not have any impact or association on the outcomes given in Figure 3. Earlier studies have reported that the association between adherence and reduced injuries is biased.

We understand the concern of the reviewer and also the lack of clarity in the initial manuscript. In line with the previous answer, Figure 3 represents the association of between IMAS (person-specific medication-taking behaviour) and incidence and not the association between one certain drug and occurrence of disease. IMAS is a proxy variable to overall health-behaviour which in turn predicts the incidence of some diseases. We hope the amendments in the manuscript have clarified this idea.

Reviewer #3 (Remarks to the Author):

The study uses a robust methodology, consistent selection of chronic-use active pharmaceutical ingredients, and advanced statistical modeling. The concept that medication-taking behavior is a stable, individual-level trait that transcends therapeutic classes is novel and impactful. The idea of using IMAS as a predictor for adherence and health outcomes may represent a step forward in personalized medicine.

The results are of immediate relevance to both pharmacy and broader health research. For pharmacists, it offers a tool (IMAS) for identifying patients at risk of nonadherence across multiple medications; For policymakers, it demonstrates how real-world prescription data can support risk stratification and targeted interventions; and for Public health impact, it proposes a scalable, data-driven framework to understand adherence in the population.

Some comments:

1. Throughout the manuscript, the authors use the term "ingredient" to refer to the pharmacologically active component of a medicinal product. However, the correct and internationally accepted terminology in pharmaceutical sciences, particularly under the European Medicines Agency (EMA) and International Council for Harmonisation (ICH) guidelines, is "active substance" or "active pharmaceutical ingredient (API)." Most peer-reviewed publications on medication adherence and pharmacoepidemiology use "active substance" or "API." Consistency

with this terminology facilitates better indexing, searchability, and comprehension across disciplines.

Thank you for pointing this out. We have replaced “ingredient” with “active substance” throughout the manuscript.

2. I recommend that the authors briefly explain what CMA5 entails and what the AdhereR package does, ideally with one or two sentences summarising its key assumptions. This will improve accessibility of the methods section for interdisciplinary readers, particularly clinicians, pharmacists, and public health professionals who may not be familiar with adherence algorithms implemented in R.

We have added short explanation concerning the CMA5 and AdhereR package to the Methods section as recommended.

3. In addition, please explicitly state whether CMA5 was calculated using dispensing data only, or if prescription issuance was also factored in. Based on current phrasing, it appears that dispensed prescriptions were used, but this should be clearly confirmed. In some parts of the manuscript (e.g., methods section), the text refers to “prescribed medications,” while in others it mentions “purchased prescriptions” or the use of pharmacy claims data. This could create confusion for readers regarding the true basis of the adherence calculations.

Thanks for pointing out this discrepancy. Only purchased prescriptions have been included into the analysis. We have look through the whole manuscript to improved consistency and clarity.

4. The concept of IMAS is central to the manuscript and one of its most innovative contributions. However, its definition, purpose, and interpretability are not clearly explained for a broader audience. I recommend: Introducing IMAS more clearly and accessibly in the Introduction, with a concise definition in plain language; Including a brief conceptual summary in the Results section alongside the statistical explanation; Adding a sentence to the Abstract highlighting IMAS as a novel person-level metric that predicts both adherence and health outcomes.

We have amended the whole manuscript according to these recommendations and tried to explain IMAS and the results concerning IMAS more thoroughly and in more plain language. We also added the Plain language summary section that is recommended by the journal.

I also recommend to address the following additional comments:

5. 38% of prescriptions lacked information on days of supply and were imputed. Though addressed via fixed effects and ingredient-level modeling, this could introduce medication-specific biases?

We would argue that the risk of medication-specific bias is modest as the proportion of imputed values was rather similar between medications. During the review period, we have published a preprint where the imputation and its effect is described in more detail (Malk et al, 2025). This analysis also showed that the missing information was not medication specific. Most of the missing values of day’s supply originated from 2012–2016. These were the first years of implementing the digital prescription system and daily dose information was not compulsory for the doctors. After improvement of the system when more input fields were made compulsory, the proportion of missing values reduced significantly. We have modified the wording concerning the limitations of day’s supply values to better explain applied approach and its possible effect on the outcome.

6. Adherence was estimated from refill behavior (? see above), not actual consumption. This is a common limitation, but it should be highlighted more clearly, at least in the discussion.

Yes, adherence was estimated based on refill behaviour and no information about actual consumption is available. We have addressed this issue in more detail in the limitations paragraph in Discussion section.

7. While IMAS shows year-over-year stability, the potential confounding effects of acute health events or life changes on adherence trajectories are mentioned but not modeled directly?

Yes, potential confounding effects of acute health events or life changes trajectories in currently not modelled directly and this is one topic for future studies.